

# High Efficiency Configuration Space Sampling
# – probing the distribution of available states

Paweł T. Jochym ⬛ * and Jan Łażewski ⬛

Institute of Nuclear Physics, Polish Academy of Sciences, Cracow, Poland

* pawel.jochym@ifj.edu.pl

## Abstract

Substantial acceleration of research and more efficient utilization of resources can be achieved in modelling investigated phenomena by identifying the limits of system's accessible states instead of tracing the trajectory of its evolution. The proposed strategy uses the Metropolis-Hastings Monte-Carlo sampling of the configuration space probability distribution coupled with physically-motivated prior probability distribution. We demonstrate this general idea by presenting a high performance method of generating configurations for lattice dynamics and other computational solid state physics calculations corresponding to non-zero temperatures. In contrast to the methods based on molecular dynamics, where only a small fraction of obtained data is used, the proposed scheme is distinguished by a considerably higher, reaching even 80%, acceptance ratio and much lower amount of computation required to obtain adequate sampling of the system in thermal equilibrium at non-zero temperature.

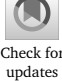

# 1   Introduction

Every system can be successfully studied by methodical observation of its behaviour for a long enough time. However, especially for slowly changing characteristics, this could take proverbial eons. On the other hand, some elementary knowledge of possible features and existing constrains allows one to limit available states of the studied system and determine the probability distribution of these states in the configuration space. As a result, the system can be modeled based on its probable configurations. To illustrate this idea, we present its application to studies of vibrational properties of solids.

A number of problems in solid state physics connected with lattice dynamics can be effectively addressed with inter-atomic potential models constructed using data obtained from quantum mechanical calculations (e.g. Density Functional Theory – DFT). Probably the simplest of such models is harmonic approximation developed by Born and von Kármán at the beginning of the 20th century [1–3]. Over the years multiple increasingly more sophisticated models have been developed: Quasi-Harmonic approximation (QHA) [4], Temperature-Dependent Effective Potential [5–7], Self-Consistent Phonons (SCPH) [8] or Parlinski's approach [9], to name just a few. All the above mentioned schemes share common feature – they need an appropriate set of data to build a model of inter-atomic potential which is essential for this type of methods. The data set should correspond to the system at thermal equilibrium or other physical state. It is usually comprised of atomic positions as well as resulting energies and forces calculated with some quantum mechanical (e.g. DFT) or even effective potential method.

Presently, molecular dynamics is often used to investigate systems at non-zero temperature in thermal equilibrium. This is done either directly – by analysis of the MD trajectory – or as a source of configurations for building the mentioned effective models of the inter-atomic potential to be used in further analysis (e.g. with programs like ALAMODE [10,11] or TDEP [7]). Both cases involve a very costly stage of running long MD calculations [12]. Since uncorrelated configurations from different parts of the phase space are required, they are generated by appropriate spacing of the sampling points over the computed trajectory or even by performing multiple independent MD runs. At the end only a small fraction of calculated configurations is used (typically $1-10\%$). Therefore, using MD in this context is exceedingly wasteful. This makes it not only very expensive but also useless for larger and more complicated systems (of hundreds or more atoms), where even static, single-point DFT calculations are challenging. In such cases running thousands of MD steps becomes prohibitively expensive and impractical.

In this work we propose a new, High Efficiency Configuration Space Sampling (HECSS) method for modelling systems in non-zero temperature, including non-harmonic effects, without using MD trajectory. We also indicate its possible application to some additional cases like disordered systems or large, complicated systems.

# 2   General idea of HECSS

To reproduce the thermal equilibrium in the system, independent configurations of displacements consistent with a desired non-zero temperature should be selected. Having any initial approximations for the lattice dynamics of the system (e.g. standard harmonic approach [2,4,13]) one can estimate temperature-dependent atomic mean-square-displacements (MSD) from a small set of force-displacement relations. Using these MSD data as a first approximation, the atomic displacements with normal distribution around equilibrium positions can be easily generated. There is, however, a subtle issue around displacements generated this way – they are *uncorrelated* between atoms, while in reality atomic displacements are correlated

at least with their close neighbours. For example, it is easy to see that a simultaneous out-of-phase movement of neighbouring atoms towards or away from each other will generate larger changes in energy than a synchronous in-phase movement of the same atoms. The former configuration should be represented with lower probability than the later, instead of equal probability present in the above simplistic scheme. Thus, while the static configurations generation may be a correct direction in general, such a naive approach is not sufficient. One can see that some additional mechanism is required to adjust probability distribution of generated samples in order to accurately reproduce configurations drawn from thermodynamic equilibrium ensemble. Classical statistical mechanics points to such a scheme for selection of configurations representing a system in thermal equilibrium.

The general form of the equipartition theorem says that a generalized virial for any phase space coordinate (i.e. generalized coordinate or momentum) is proportional to temperature when it is averaged over the whole ensemble:

$$\left\langle x_m \frac{\partial H}{\partial x_n} \right\rangle = \delta_{mn} k_B T \,, \tag{1}$$

where: $x_n$ – generalized coordinate or momentum, $H$ – Hamiltonian, $T$ – temperature, $k_B$ – Boltzmann's constant and $\delta_{mn}$ – Kronecker's delta. If we assume ergodicity of the system, the ensemble average may be replaced with time average. For momenta this leads to the average kinetic energy per degree of freedom being equal to $k_B T/2$ and provides the kinetic definition of temperature. However, the relation holds also for derivatives of Hamiltonian with respect to positions. Considering relation (1) for a *single* atomic displacement from the equilibrium configuration described by coordinate $q$, and assuming the potential energy depends only on position, we can write position-dependent part of the Hamiltonian (i.e. the potential energy $E_p(q)$) as a Taylor's expansion with respect to the atomic displacement $q$ from the equilibrium configuration:

$$E_p(q) = \sum_{n=2}^{\infty} C_n q^n \,, \tag{2}$$

where the expansion coefficients $C_n$ are, in general, functions of all remaining coordinates (displacements). Note that, this is *not* a general, multi-dimensional, polynomial expansion – just a single coordinate expansion required by the equipartition theorem (1), which now takes the form:

$$k_B T = \left\langle q \sum_{n=2}^{\infty} n C_n q^{n-1} \right\rangle = \sum_{n=2}^{\infty} n C_n \langle q^n \rangle \tag{3}$$

and if we write $n$ as $(n-2)+2$ and divide both sides by 2 we get:

$$\left\langle E_p(q) \right\rangle = \frac{k_B T}{2} - \sum_{n=3}^{\infty} \frac{n-2}{2} C_n \langle q^n \rangle \,, \tag{4}$$

which is similar to the kinetic energy counterpart except for an additional term generated by the anharmonic part of the potential and defined by the third and higher central moments of the probability distribution of the displacements. If we can assume that the second term of the Eq. 4 is small in comparison with $k_B T$, we get a formula for the average potential energy of the system. Note that for harmonic systems the second part vanishes. For anharmonic systems omission of higher terms in Eq. 4 will provide first-order approximation of the mean potential energy. Considering the quality and applicability range of this approximation, one should note that substantial higher-order terms are present only in parts of the formula connected with strongly anharmonic modes. Furthermore, for every atom in centro-symmetric position all odd-power moments vanish and the first non-zero moment is the fourth one. In addition,

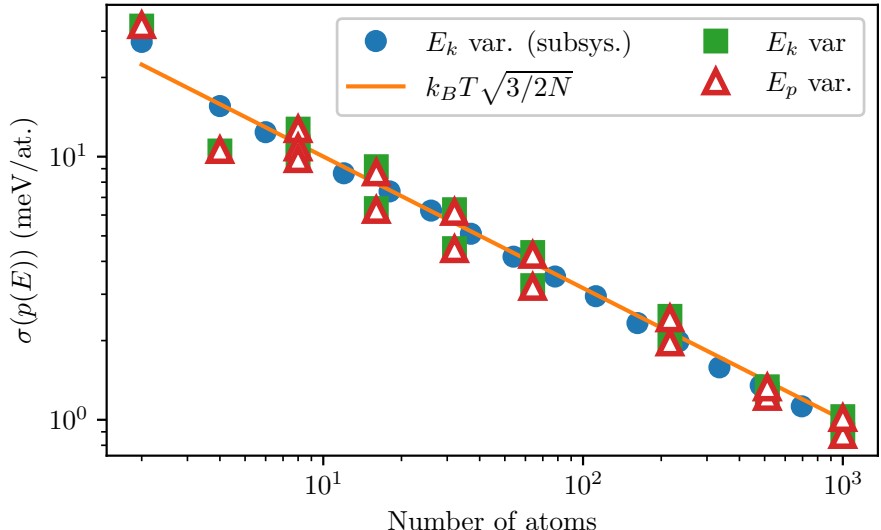

Figure 1: Variance of the energy distribution as a function of system size compared with prediction of the central limit theorem (orange line). Results for different numbers of randomly chosen coordinates of $5 \times 5 \times 5$ system (blue circles) were put together with variance of both the kinetic (green squares) and potential (red triangles) energies of smaller systems (defined in the text). Data extracted from MD run at $T = 300$ K.

the main effect of the second term in Eq. 4 can be understood as correction to the temperature scale of the modeled system – not the qualitative change of the energy distribution. This correction may be estimated by, for instance, deriving the $C_n$ coefficients from the force constants matrices determined in phonon calculation. Finally, the formula for the potential energy of the whole system contains similar terms for all modes. Judging by extremely high efficiency of harmonic approximation for crystal lattice dynamics, we can expect that this averaging will make proposed approximation effective for a wide range of systems. On the other hand it is this additional term in potential energy where all non-harmonic physics resides and it indicates the most important limitation of the proposed method: conditions where energy variance is divergent (i.e. phase transitions with divergent heat capacity). As long as the energy variance is stable the non-harmonic effects should be limited to the temperature re-calibration and adjustment of the distribution variance. These adjustments should be considered as a next-level corrections to the presented formulation and subject of future research.

To sum up, MD provides a representation of the system with the properly distributed kinetic energy. For a single particle it is a Maxwell-Boltzmann distribution. By virtue of the central limit theorem (CLT) [14,15], if we increase the number of particles we will approach at infinity (i.e. in the thermodynamical limit) a Gaussian distribution with the same average (the same mean) and the variance which is scaled as inverse number of particles. As we can see for kinetic energy the relation is very simple whereas for the potential energy we have a quantity approximately close to temperature if the system is not too far from a harmonic one. Nevertheless, we do not know, in general, the form of the distribution of the potential energy. That constitutes substantial difficulty, which fortunately can be overcome by application of the CLT to calculate distribution of potential energy.

The CLT states that for any reasonable probability distribution, the distribution of the mean of the sample of the independent random variable drawn from it, tends to the normal distribution with the same mean and variance scaled by the square root of the number of samples.

The *reasonable* class is fairly broad here, including many physically interesting cases by virtue of requiring only a finite variance and a well-defined mean. Obviously, this excludes important case of systems close to phase transitions with divergent specific heat (i.e. divergent energy variance, e.g. melting). Thus, for potential energy per degree of freedom we can expect the probability distribution to asymptotically converge to the normal distribution:

$$\sqrt{3N}\left(\frac{1}{N}\sum_i E_i - \langle E \rangle\right) \xrightarrow{d} \mathcal{N}(0, \sigma). \tag{5}$$

As shown above, one can approximate the $\langle E_p \rangle$ with the first term of Eq. 4 and the only unknown parameter in this formula is the variance of the distribution. Note that above expression is *independent* from the particular shape of the potential energy probability distribution for the single degree of freedom except of its mean $\langle E_p \rangle$ and variance $\sigma$. The mean is set by Eq. 4 while variance is determined by the energy conservation and the fact that total energy is a sum of potential and kinetic energy – thus their variances should match, as Fig. 1 clearly illustrates.

However, we should keep in mind that the Eq. 5 is true *asymptotically*. And for that reason we need to check if this relation has any practical use for *finite*, and preferably not too large, $N$. The common wisdom in statistical community, based on Berry-Esseen theorem [16, 17], states that for $N$ above $\approx 50$ the distribution of the average is practically indistinguishable from the true normal distribution, and even for smaller $N$, if the starting distribution is not too far from normal (e.g. Maxwell-Boltzmann, uniform in range, triangular, close to symmetric), the convergence is usually very quick ($N \approx 15-20$). The hard bound from the theorem is for the supremum norm of the difference between the cumulative distribution of the average and normal cumulative distribution to be less than $L_p/\sigma^3\sqrt{N}$, where $L_p$ is a number proportional (with constant $\approx 1$) to the expectation value of the third moment of the absolute value of the random variable.

## 3 Sampling of probability distribution

To verify if the mentioned heuristic rule holds true for the typical kinetic and potential energy distributions, we have checked this hypothesis against actual MD data of a typical system. This test does not require high-accuracy forces and energies but demands ability to efficiently calculate moderately sized systems (e.g. 1000 atoms). Thus, instead of using DFT as a source of energies/forces we have used effective potential model of the cubic 3C-SiC crystal. We have used LAMMPS [18] implementation of the Tersoff potential with parameters derived in [19,20] and the NVT-MD implemented in ASAP3 module of the Atomic Simulation Environment (ASE) [21]. High performance of this implementation allowed for $5 \cdot 10^4$ time steps (of 1 fs length) runs of the $5 \times 5 \times 5$ supercell (1000 atoms) to be executed on a single 8-core server in just a few hours.

The kinetic and potential energy probability distributions extracted from MD runs of systems of 2, 8 and 64 atoms (i.e. 6, 24, 192 degrees of freedom) are presented in Fig. 2. At this stage we are interested in the speed of convergence of the probability distribution, and this experiment shows that for typical distributions present in crystals (i.e. $\chi^2$, Maxwell-Boltzmann) the convergence is indeed fairly quick. Already at the $N_{DOF} = 24$ (8 atoms, central column in Fig. 2) the deviation from the normal distribution is smaller than $0.3\sigma$ discrepancy in position of the mode (maximum) and mean, at $N_{DOF} = 192$ (i.e. 64 atoms, right column in Fig. 2) this discrepancy drops below $0.1\sigma$. The difference between median and mean drops below $0.1\sigma$ at 8 atoms already. The results in Fig. 2 illustrate that the approximation of probability



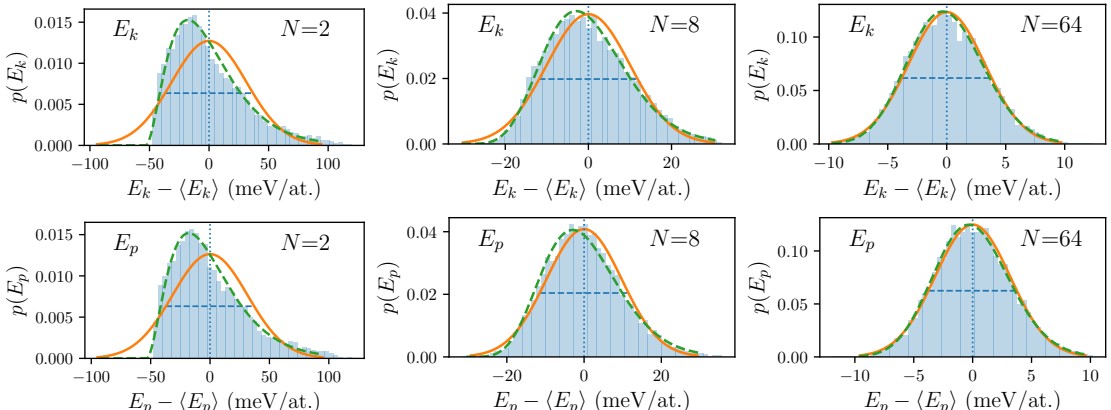

Figure 2: Probability distribution for single atom kinetic (upper) and potential (lower) energies averaged over $N = 2$, 8 and 64 randomly selected atoms. Solid orange lines show fitted normal distributions while dashed green lines show $\chi^2$ distribution for $3N$ degrees of freedom fitted to energy histograms. Data derived from the MD trajectory run at 300 K temperature.

distribution by normal distribution holds true equally well for distributions of the kinetic and potential energy.

This simple example demonstrates that for our practical purposes we can expect the energy distribution in crystals to follow central limit theorem predictions above $\approx 30$ degrees of freedom, for both the kinetic and potential energies. Thus, we can apply this approach even for very moderately sized systems of $10-20$ atoms.

The energy distributions in Fig. 2, derived from the MD runs mentioned above, show clearly distributions close to Gaussian for both the kinetic and potential energies even for $N_{DOF} = 24$ degrees of freedom. Furthermore, the variance of these distributions plotted against the system's size (Fig. 1) follows closely CLT prediction of Eq. 5 for parts of a larger system (blue circles in Fig. 1) as well as for the whole smaller crystals (squares and triangles in Fig. 1). The dispersion of small systems' data in Fig. 1 is due to large temperature fluctuations in small sets of particles.

Thus, we have checked that, at least in our test case, the convergence to thermodynamic and CLT limits required by the Eqs 4 and 5 is quick enough to be useful in practical calculations for systems of just tens of atoms. The main problem now is that there is no direct access to potential energy and there is no way to invert relation from positions to potential energy – even in principle – since the relation is many-to-one. Our goal here is to reproduce the potential energy distribution described by Eq. 5 and present in MD data by intelligently sampling the configuration space of the system – since this is the only input we can directly specify. Fortunately, computational statistics provides multiple algorithms dedicated to the task of sampling of indirectly specified probability distributions. In particular, the Metropolis-Hastings Monte-Carlo [22, 23] seems well suited to our purposes. To use it effectively we need to generate a prior distribution which covers the domain and, preferably, is fairly close to the target distribution. Obviously, we are unable to generate configurations corresponding to the distribution from Eq. 5 but we can use physically motivated approximation. We propose to approximate displacements of atoms in the system by Gaussian probability distribution with variance tuned to the temperature and to the resulting energy. Our HECSS software package provides the Metropolis-Hastings implementation together with a tuned prior probability distribution generator. The tuning algorithm adjusts the variance of the atomic displacement distribution in

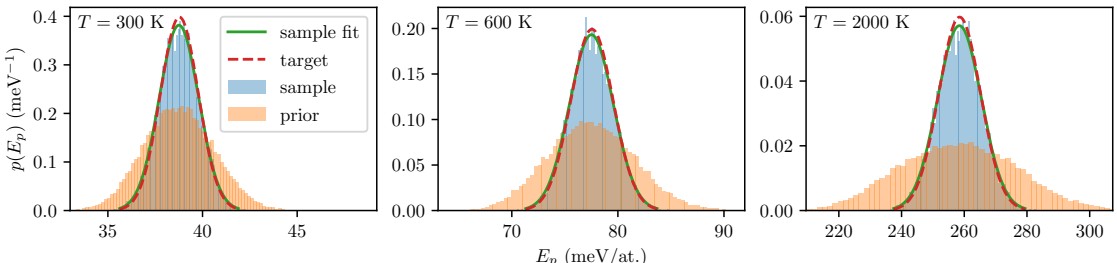

Figure 3: Prior energy probability distribution (orange filling) versus target distribution (blue filling). The lines indicate target distribution (red dashed line) and Gaussian distribution fitted to generated sample (green solid line). The data was generated for temperatures $T = 300$, $600$ and $2000$ K with $\delta = 0.1$ in HECSS procedure (see description in the text).

each step: $\sigma_{n+1} = (1 + s(E_p(x_n)))\sigma_n$, according to the modified logistic sigmoid function:

$$s(E_p) = \delta \cdot \left( \frac{2}{1 + e^{-(E_p - E_0)/(w \cdot \sigma_{E_p})}} - 1 \right), \tag{6}$$

where $\sigma_{E_p} = k_B T \sqrt{3/2N}$ is the variance of the target potential energy distribution (5) and $\delta \approx 0.005 - 0.02$ is a small tuning parameter controlling the speed of the variance adjustment, while $w \approx 3$ controls the width of the prior distribution. Both parameters have substantial practical importance – they influence the effectiveness of the procedure – but play no fundamental role in the algorithm. Changing these parameters to the unsuitable values leads only to slower convergence of the procedure, since the Metropolis-Hastings algorithm produces asymptotically the target distribution from any prior distribution non-vanishing over the domain [23]. The prior distribution we are proposing here is already of similar shape to the target one and it includes a parameter self-tuning algorithm. Thus, it needs only several additional samples at the start of the procedure to properly tune the width parameter – if it was not set correctly. Our selection of the prior distribution means getting higher than 50% (in practice even above 80%) acceptance ratio instead of a few percent or even less if the prior distribution was very far from the target. The typical good relationship between prior and target distribution as well as the sampling produced by the proposed algorithm is illustrated in Fig. 3. The data in this figure has been generated with the artificially large $\delta$ parameter (0.1 instead of typical 0.005 − 0.02) – to make the difference between prior and posterior distribution more obvious. Such a large $\delta$ makes no difference in the posterior distribution but substantially lowers the acceptance ratio (usually below 50%). All remaining data has been generated with typical $\delta = 0.01$.

The near-independent drawing of each step in the algorithm means that each sample from the produced set is potentially usable. Therefore, the burn-in period may be reduced to just a few samples required for tuning of the prior distribution parameters. The only source of possible correlations between samples in consecutive steps is the change in variance of the prior distribution, which is tuned after each step according to the sigmoid function (defined by Eq. 6). This is a very weak correlation since the variance is not supposed to change by more than $\delta \approx 0.5 - 2\%$. What is more, these parameters seem to be independent from the size of the system and their values appear not critical, judging from our experience. This property stems from the fact that the interatomic forces are only slightly modified by the transition from the small to large supercell while the average displacement is determined mostly by the overall shape of the interatomic potential. The variance of the prior distribution, which is self-tuning, should be estimated within 20% accuracy to limit the required burn-in period to

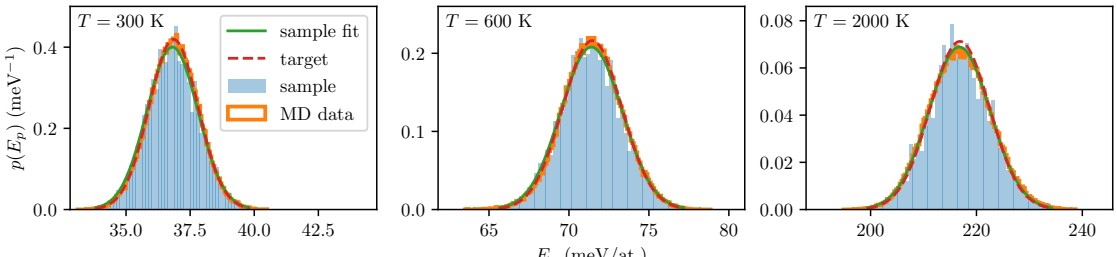

Figure 4: Probability distribution of potential energy per atom generated with HECSS scheme (blue shape) versus distribution extracted from the MD trajectory (orange contour). The dashed line indicates normal distribution fitted to MD sample. The data corresponds to the temperatures $T = 300$, 600 and 2000 K (as labeled in sub-figures).

just one or two samples. Thus, the initial tuning may be performed using a small supercell or even a primitive unit cell – depending on the system – by just recording the self-tuning trajectory of the algorithm and using final tuned parameters as their initial values in the production run. The possible correlations introduced in the HECSS generated data result only from the fact that if the $n$-th sample leads to exceptionally small or large energy, the next sample is drawn from the positional distribution with variance increased or reduced, respectively, by a small amount (no more than $(1 + \delta)$ times in extreme cases). Thus, the probability of larger energy following the exceptionally small energy in the sampling chain (or a smaller sample following an exceptionally large one) is slightly increased. Note, however, that this does not introduce any correlations in any particular coordinate. On the other hand, in the MD trajectory the correlations arise from the non-random character of the particle trajectory. The output of proposed algorithm is a series of samples (i.e. configurations) which reproduce expected probability distribution (5) of potential energy for the system in thermal equilibrium at the target temperature. The comparison between the potential energy probability distribution in the samples generated by HECSS and extracted from the MD run is depicted in Fig. 4.

## 4 Convergence of derived quantities

The results presented above demonstrate that it is possible to effectively generate samples with potential energy distributions consistent with the data from the MD trajectories. The remaining, much more difficult, question is whether these samplings indeed provide an appropriate representation of the system in thermal equilibrium at a given temperature. This issue may be tested in various ways. In this work we propose to check if the potential model built basing on the HECSS-generated displacement-force data provides phonon frequencies and lifetimes consistent with those derived from the MD trajectory data.

Therefore, we have compared the results of both methods (i.e. MD and HECSS) obtained from the calculations of 3C-SiC crystal with LAMMPS potential used in the previous section. The samples generated by both methods have been used to build force constants matrices for the material with ALAMODE program. The calculations were performed using $5 \times 5 \times 5$ supercell and the reciprocal space integrations required for phonon lifetimes were executed over $20 \times 20 \times 20$ grid in the reciprocal space. The model used second- and third-order force constants determined by fitting displacement-force relationship to the data sets containing varying number of samples.

The resulting phonon frequencies derived from harmonic components and lifetimes ex-

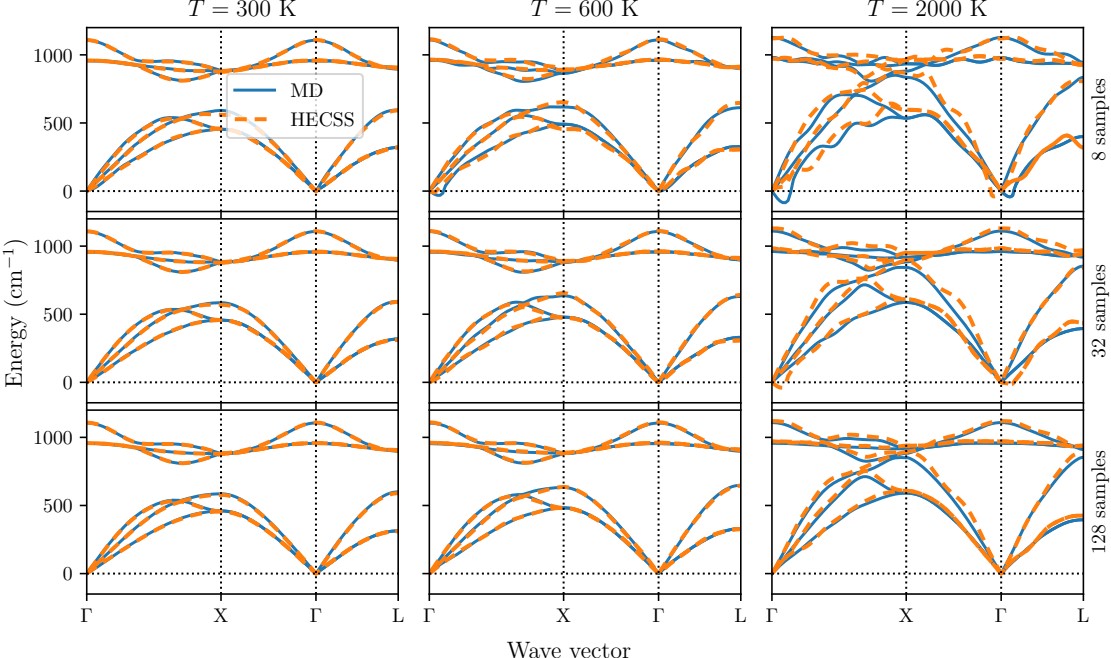

Figure 5: Consistency and convergence of phonon frequencies in 3C-SiC crystal determined with harmonic model derived from MD (solid lines) and HECSS (dashed lines) data. The plots correspond to the temperatures $T = 300$, 600 and 2000 K. The rows illustrate convergence of the result with number of samples (8, 32 and 128).

tracted from third order coefficients of the same model are presented in Fig. 5 and Fig. 6, respectively. These findings demonstrate not only high-level of consistency between both data sets and models, but also similar convergence characteristics between both methods. The Figs 5 and 6 show the results calculated at three temperatures (300, 600 and 2000 K for phonon frequencies and 100, 300 and 600 K for phonon lifetimes) and several sizes of the data set (8, 32 and 128 for frequencies, 16 and 128 for lifetimes). Both figures clearly demonstrate that the agreement and convergence of the results derived from both methods is very good for low and moderate temperatures (up to 600 K for frequencies and 300 K for lifetimes) and remains reasonably good for higher temperatures (even 2000 K for frequencies – Fig. 5). The RMS difference between frequencies obtained from the 128 samples (lower row in Fig. 5) are: 2.8 cm$^{-1}$ for $T = 300$ K, 4.5 cm$^{-1}$ for $T = 600$ K and 22.5 cm$^{-1}$ for $T = 2000$ K. It should be noted that for higher temperatures the size of the data set needs to be substantially increased ($2-4$ times) comparing to the size sufficient for convergence at low temperatures. It is worth noting that the last column in Fig. 5 ($T = 2000$ K) shows no significant difference in convergence characteristics between data obtained from MD and HECSS procedures.

The higher order properties are more difficult to derive at high temperatures (600 K, Fig. 6) – which can be expected. This may be an intrinsic property of the proposed procedure or may be caused by insufficient accuracy of the LAMMPS potential used – which is not optimized for this type of calculation, especially at high temperatures. It should be noted that derivation of phonon lifetimes is very sensitive to the accuracy of the interaction model. This issue should be investigated in the future research, preferably using high-accuracy DFT-based calculation for energy and force determination. Nevertheless, the data presented in Fig. 6 demonstrates remarkable agreement between phonon lifetimes calculated with both methods for small and moderate temperatures (100 and 300 K). The agreement which holds over four orders of magnitude. Furthermore, the data for $T = 600$ K (right column in Fig. 6) illustrates that

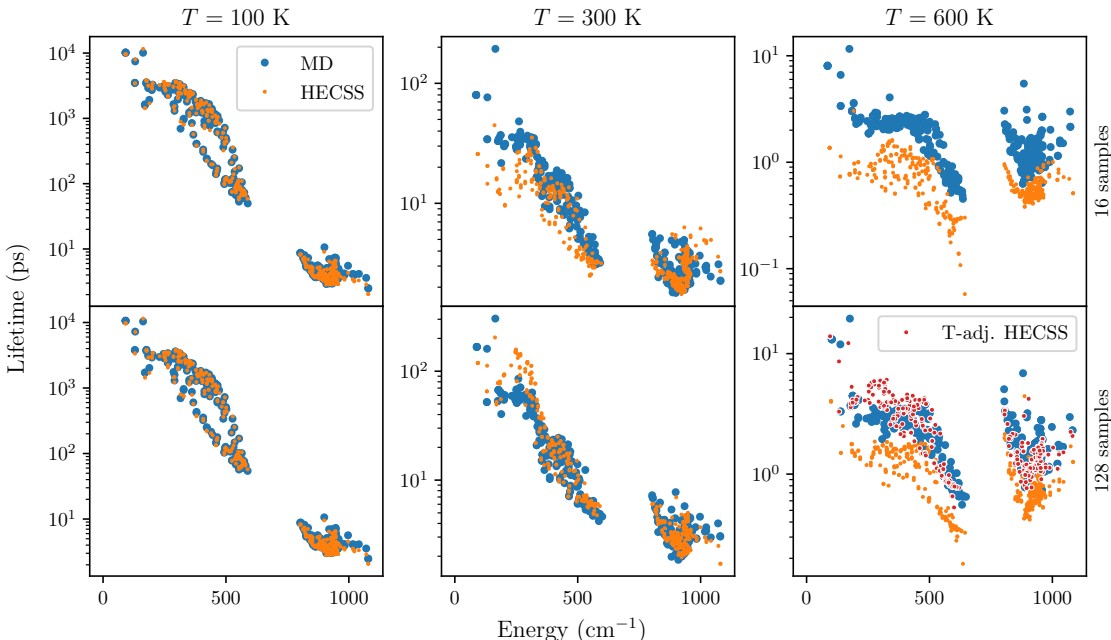

Figure 6: Consistency and convergence of 3C-SiC phonon lifetimes derived with third order model based on MD (blue) and HECSS (orange and red) generated data. The data corresponds to three temperatures $T = 100$, 300 and 600 K. Two sample sizes used are: 16 – enough to obtain well-converged phonon frequencies (see Fig. 5) and 128 – selected after convergence testing of the obtained lifetimes. The red dots indicate lifetimes obtained by lowering the temperature in HECSS procedure by 90 K.

even at higher temperatures most of the accuracy can be recovered by simple temperature scaling mentioned in Section 2. The red points in lower panel ($T = 600$ K, 128 samples) are obtained from the HECSS procedure executed at the temperature lowered by 90 K. Their better agreement with MD-derived results (blue dots) indicates that this "temperature calibration" effect may indeed be a leading next-order correction to the proposed procedure. The derivation of more sophisticated, higher order corrections for the high temperature regime is a promising direction of future research in this area.

It is important to note that HECSS-generated data sets consist of first $N$ drawn samples (after initial burn-in period of 3 samples), not the $N$ samples selected from the larger set, as it is done with MD trajectory. Obviously, if one was forced to run as many steps of HECSS algorithm as time steps of the MD trajectory the whole effort would be pointless. The experience gained during the development of the algorithm indicates that a set of $N$ configurations based on DFT energy calculation can be generated in time equivalent to approximately $2*(N+10)$ time steps of MD – which is not enough to generate even single well-thermalized sample for $N < 500$. It is evident that the results of both approaches are very similar, despite a large difference in necessary computational effort – which provides a clear justification for future application of the presented method to the much more expensive DFT-based variant of the potential energy calculation.

## 5 Conclusions

We have introduced a new high efficiency configuration space sampling (HECSS) scheme as an alternative for application of Molecular Dynamics as a source of configurations represent-

ing systems at non-zero temperatures. The results presented above demonstrate potential of the proposed HECSS method to generate faithful configuration samplings for systems in thermal equilibrium, which can be used to investigate anharmonic effects present in crystalline solids. It is worth noting that this method is not limited to crystals or to only geometric degrees of freedom. In principle, it is possible to extend its applicability to magnetic degrees of freedom or disordered systems. Furthermore, due to its inherent ability to provide 3×number-of-atoms force-displacement data points per configuration, it reduces number of energy/force calculations required for simple harmonic model determination. This reduction is much more pronounced in higher-order models, where number of independent variables is usually large. It should also be emphasized that the generated samples are drawn from the physically meaningful distribution and not from the non-physical, single axis displacements. This difference may become important if there is any substantial anharmonicity in the system, which couples degrees of freedom. While the proposed approach is demonstrated above on lattice dynamics calculation its potential applicability is not limited to this field – it may be used in other cases where the set of configurations corresponding to thermal equilibrium is required.

## Acknowledgments

The authors would like to express their gratitude to Krzysztof Parlinski, Przemysław Piekarz, Andrzej M. Oleś and Małgorzata Sternik for very inspiring and fruitful discussions. This work was partially supported by National Science Centre (NCN, Poland) under grant UMO-2017/25/B/ST3/02586.

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
