# Peer review of "High Efficiency Configuration Space Sampling -- probing the distribution of available states"

_SciPost Physics, doi:SciPost Phys. 10, 129 (2021)_

## Round 1 · Referee Report · Anonymous (Referee 1) · 2021-2-26

Report

The authors of the manuscript "High Efficiency Configuration Space Sampling – probing the distribution of available states" present a configuration sampling approach based on the Metropolis-Hastings algorithm.
The authors claim that their approach is more efficient than other established approaches based on molecular dynamics (MD) propagation. They validate their numerical approach by computing the phonon dispersion and lifetimes
of the cubic 3C-SiC crystal and comparing against MD benchmark results.

I have found a few gaps in the presentation which prevent a full grasp of their assumptions and numerical validation. Their arguments and derivations cannot be fully reproduced by qualified experts

For instance, the authors claim they sampled at finite temperature, but the value of the temperature is not provided anywhere in the text.

If the temperature is much smaller than the melting temperature (~3,000 K for SiC), the harmonic approximation is expected to be quite accurate and the the Gaussian "prior" sampling described in Sec. 3 should be almost exact. This seems to agree with the large acceptance ratio (up to 80%) observed by the authors.

A very high acceptance ratio is not necessarily an advantage of the approach as it may imply large correlation in the sample. The authors should discuss the position autocorrelation function obtained from their approach and compare against the one obtained using canonical MD sampling. The best acceptance rate is the one which minimises the autocorrelation time.

The agreement between the Monte Carlo and MD phonon dispersion shown in Fig. 5 is to be expected if the anharmonicity is negligible. The agreement on the phonon lifetimes is a tougher check, but it is hard to draw any quantitative conclusion from Fig. 6. The points are pretty scattered over a semi-log plot, which means that the error can be rather large.
The authors should discuss a more quantitative estimator, e.g., the square root of the sum over the wave-vectors and bands of the square deviation of the Monte Carlo and MD phonon lifetimes.

Lines 34-35: The authors mention "running a 30000 steps MD". Why exactly this number of steps?

Eq. (1): Not all the symbols have been introduced in the text.

Eq. (2): The authors implicitly assume a two-body force field. What would happen in the case of a many-body force field like the embedded atom model or Tersoff potentials?

Lines 80-81: The sentence "Only experience can tell us how good this approximation is and how wide its applicability range is" is not correct and underrate the role of a large body of numerical analysis.

Lines 102-103: The sentence "The reasonable class is very broad here, certainly containing all physically interesting cases by virtue of requiring a finite variance and a well-defined mean" is not correct, unless phase transitions are excluded. The variance of several quantities diverges close to a phase transition.

Eq. (5): The equal sign is not correct as for N\to\infty the variance of the distribution of the right-hand side is zero.

Line 125: A "single server" is not a well-defined object: how many CPU's were used?

In comparing the computational efficiency of MD and Monte Carlo methods, one has to take into consideration the time spent generating random numbers, which may be more computationally demanding than propagating a trajectory, given the same number of force-field evaluations.

In conclusion, I do not feel comfortable in suggesting the manuscript in its current form.
  • validity: -
  • significance: -
  • originality: -
  • clarity: -
  • formatting: -
  • grammar: -

Author:  Paweł Jochym  on 2021-03-04  [id 1286]

(in reply to Report 1 on 2021-02-26)

We thank the referee for reading our paper and spotting omissions and mistakes in our text. We would like to immediately address the referee's comments and correct mistakes pointed in his report. Naturally, we intend to include these corrections in the resubmitted manuscript. We hope that the following clarifications will enable full understanding of our approach. We address the comments paragraph by paragraph quoting the referee before our response.

I have found a few gaps in the presentation which prevent a full grasp of their assumptions and numerical validation. Their arguments and derivations cannot be fully reproduced by qualified experts For instance, the authors claim they sampled at finite temperature, but the value of the temperature is not provided anywhere in the text.

The text was intended as a demonstration of the proposed method not as a reproduction or prediction of a particular experimental result. This may perhaps explain unfortunate omission of sampled temperature in the text. While the temperature can in fact be extracted from the average energy in Fig. 3 and 4, it should also be stated in the text explicitly. The sampling temperature for all presented data is 300 K, chosen as standard ambient temperature. However, we have tested our algorithm for a number of other temperatures, up to 2000 K and there was no qualitative difference in its performance or properties.

If the temperature is much smaller than the melting temperature (~3,000 K for SiC), the harmonic approximation is expected to be quite accurate and the the Gaussian "prior" sampling described in Sec. 3 should be almost exact. This seems to agree with the large acceptance ratio (up to 80%) observed by the authors.

The shape of the energy distribution is not determined by harmonicity of the potentials but by the Central Limit Theorem and the size of the system. The difference between prior distribution (which comes from our approximation of the displacement distribution) and the target distribution does not stem from the anharmonicity of the potential but mainly from the fact that the displacements of the atoms in the crystal are not independent (clearly stated in our text). Please note that we have no direct access to the potential energy of the system - we can only specify the geometry and calculate the resulting energy. Thus, we have no ability to directly generate the target energy distribution from Fig. 3. The high acceptance ratio comes from our selection of the displacement distribution and tuning algorithm described in the paper. The Metropolis-Hastings (M-H) algorithm can generate a target distribution from any prior which is non-zero over the domain. However, the acceptance ratio may be very low if you fail to use the prior which is a good approximation of target distribution. It is a well-known fact in the numerical statistics community that the good selection of the prior distribution is the key to an effective use of probability distribution sampling algorithms.

A very high acceptance ratio is not necessarily an advantage of the approach as it may imply large correlation in the sample. The authors should discuss the position autocorrelation function obtained from their approach and compare against the one obtained using canonical MD sampling. The best acceptance rate is the one which minimises the autocorrelation time.

The issue of sample correlation would be indeed important if we used a 'random walk'-type algorithm for the prior generation (which is a popular variant of the M-H algorithm). Instead, we use independent samples and the only possible correlation between them arises from a very small change (less than 2%) in the variance of the position distribution. We discuss this issue in the paragraph starting in line 196. On the other hand, the MD derived data has obvious autocorrelations - note that we can derive phonon frequencies from the Fourier transform of the velocity autocorrelation function along the MD trajectory. Thus, it is necessary to separate sampling points on the trajectory by substantial intervals allowing for these correlations to die out. Nevertheless, all time steps between the sampling points still need to be calculated, which leads to large inefficiency of the MD as a configuration generator. To further clarify the issue we suggest expanding the explanation in the text by the following paragraph:

'The possible correlations introduced in the HECSS generated data result only from the fact that if the n-th sample leads to exceptionally small or large energy, the next sample is drawn from the positional distribution with variance increased or reduced, respectively, by a small amount (no more than $(1+\delta)$ times in extreme cases). Thus, the probability of larger energy following the exceptionally small energy in the sampling chain (or the other way around: a smaller sample following an exceptionally large one) is slightly increased. Note, however, that this does not introduce any correlations in any particular coordinate. In the MD trajectory the correlations arise from the non-random character of the particle trajectory.'

The agreement between the Monte Carlo and MD phonon dispersion shown in Fig. 5 is to be expected if the anharmonicity is negligible. The agreement on the phonon lifetimes is a tougher check, but it is hard to draw any quantitative conclusion from Fig. 6. The points are pretty scattered over a semi-log plot, which means that the error can be rather large. The authors should discuss a more quantitative estimator, e.g., the square root of the sum over the wave-vectors and bands of the square deviation of the Monte Carlo and MD phonon lifetimes.

Indeed, for the purely harmonic system the phonon frequency test is not useful since phonon frequencies are independent from the displacement size. However, if the system considered in the paper were close to harmonic, we would expect to obtain very long phonon lifetimes (since they are infinite in the harmonic system). The data in Fig. 6 demonstrates that most of the phonon modes exhibit lifetimes below 10ps - showing non-negligible anharmonicity in the model. This fact provides justification for the validity of the phonon frequency test. The phonon lifetimes are very sensitive to the accuracy of the model. This is especially true in case of large values which indicate small deviations from harmonicity and usually carry large error bars. Unfortunately, the large range (close to two orders of magnitude) of the values of lifetimes makes the simple RMS measure of differences very misleading - since the differences at high end of the range will dominate the sum. Thus, we are going to consider using RMS of logarithms of lifetimes as a better measure of relative changes in the values obtained using MD and HECSS approaches.

Lines 34-35: The authors mention "running a 30000 steps MD". Why exactly this number of steps?

The number of steps (30 000) used in the introduction was a typical relaxation time of a long-run MD suggested by the often used "rule of thumb" in MD calculations (50 times period of typical vibrations in the system). For 3C-SiC: $f\approx 10$THz = $10^{13}$Hz $ \Rightarrow t=10^{-13}$s = 100fs; 50 * 100fs = 5ps. With 1fs time step that equals 5000 steps minimum run where we can use at most half of it for actual data (you need to provide time to obtain thermal equilibrium). If we need approx. 30 data points (as required by anharmonic calculations, see Fig. 6) and they should be separated by at least 1ps interval (at least 10 typical vibrations) we get approximately 30 000 steps. The cited number itself has no 'magical' value and results from the setup of the calculations presented in the paper. To avoid impression that the number 30 000 has any special meaning, we are going to replace the number by the phrase: "thousands of MD steps".

Eq. (1): Not all the symbols have been introduced in the text.

Eq. (1): The sentence introducing missing symbols will be added to the revised text: $x_n$ - generalized coordinate, $H$ - Hamiltonian, $T$ - temperature, $ k_B$ - Boltzmann constant, $\delta_{mn}$ - Kronecker delta

Eq. (2): The authors implicitly assume a two-body force field. What would happen in the case of a many-body force field like the embedded atom model or Tersoff potentials?

Eq. (2): We make no two-body assumption, neither implied nor explicit. The formulation of equipartition theorem, Eq. (1), explicitly concerns single coordinates (the only non-zero term due to the Kronecker delta) and makes no assumption on the form of the Hamiltonian H. The Taylor expansion, Eq. (2), is not a complete expansion in all coordinates $q$ (note the scalar $q$ symbol). It is the Taylor expansion in single coordinate with coefficients ($C_n$ - proportional to partial derivatives of energy with respect to this coordinate) which are functions of all the other coordinates in the system. Furthermore, the calculations presented in the paper use the mentioned Tersoff potential developed in refs 17, 18. We understand that due to the formulation of the surrounding text this may not be entirely clear and may confuse the reader. To avoid this we are going to add a clarifying sentence before Eq. (2).

Lines 80-81: The sentence "Only experience can tell us how good this approximation is and how wide its applicability range is" is not correct and underrate the role of a large body of numerical analysis.

The unfortunate sentence 80-81 brings nothing of importance to the text. We are going to remove it in the resubmitted text.

Lines 102-103: The sentence "The reasonable class is very broad here, certainly containing all physically interesting cases by virtue of requiring a finite variance and a well-defined mean" is not correct, unless phase transitions are excluded. The variance of several quantities diverges close to a phase transition.

Variance of several quantities is indeed divergent in some phase transitions. In cases where the transition involves divergent heat capacity this includes energy variance. Thus, our phrase :"...all physically interesting cases..." was indeed wrong. The sentence is going to be corrected. We are going to specify where we can use the described procedure and clearly state that in cases where energy variance diverges, the procedure cannot be used. We thank the referee for spotting this important fact.

Eq. (5): The equal sign is not correct as for N\to\infty the variance of the distribution of the right-hand side is zero.

The Eq. (5) was an attempt to formally write asymptotic relation of the Central Limit Theorem described in the paragraph 99-102. The CLT is indeed not a limit relation but asymptotic distribution convergence relation and the Eq. (5) should use appropriate notation for such relations as convergence in distribution:

$$ \sqrt{3N}\left(\frac{1}{N} \sum_i E_i -\langle E\rangle \right) \xrightarrow{d}\mathcal{N}(0, \sigma). $$
The mistake in notation has no consequences for the arguments and conclusions presented in the text. The Eq. (5) is going to be corrected in the revised text.

Line 125: A "single server" is not a well-defined object: how many CPU's were used?

"Single server" mentioned in line 125 was used as a rough indication of the computational effort involved in the described task. It is nothing out of ordinary: 2x4 cores CPU and 32GB RAM. This is actually a fairly under-powered and old machine, less powerful than some of newer generation laptops. The information will be added to the sentence in revised text.

In comparing the computational efficiency of MD and Monte Carlo methods, one has to take into consideration the time spent generating random numbers, which may be more computationally demanding than propagating a trajectory, given the same number of force-field evaluations.

In some cases the random number generation may be indeed fairly expensive but in the case of typical systems of tens of atoms, the energy and forces evaluation is much more time-consuming. For instance, the random number generator we have used (from SciPy.stats library) takes 180$\mu$s to generate 3000 random numbers required to create one sample for the 5x5x5 supercell of 3C-SiC. The single evaluation of energy for the same cell (1000 atoms) takes 4ms (20 times longer) using ASAP3 with OpenKIM model from our calculations. A more sophisticated interaction model is bound to be even more time-consuming. Furthermore, molecular dynamics requires calculating multiple time steps per every generated sample. Considering this facts, we maintain that the proposed HECSS approach offers substantial advantage over MD as a source of configuration data.

---

## Round 1 · Referee Report · Bjorn Wehinger (Referee 2) · 2021-3-10

Strengths

Original new approach for the study of lattice lattice dynamics at finite temperatures.

Weaknesses

Presentation should be improved.

Report

The authors of the manuscript "High Efficiency Configuration Space Sampling – probing the distribution of available states" present a new method for studying lattice dynamics at finite temperatures. Their approach is based on configuration sampling the distribution of available states using the Metropolis-Hasting algorithm with a prior probability distribution derived from harmonic lattice dynamics. The authors compute anharmonic phonon dispersions and lifetimes for 3C-SiC to validate their approach and claim high computational performance due to a large observed acceptance ratio. The idea is highly original and I expect significant impact for the study of thermal properties at finite temperatures in large crystalline systems containing many atoms per unit cells. In order to fully convince the reader and justify publication in SciPost Physics, I recommend the authors to address and clarify the following:

  1. The lattice dynamics of 3C-SiC at room temperature can be described fairly well by the harmonic approximation. It seems thus no big surprise that a prior probability distribution derived from harmonic lattice dynamics converges successfully and quickly. But how well does it work for a more anharmonic situation? Although the chosen potentials might not be accurate close to melting it would be a very nice illustration to compare the dispersions and lifetimes to molecular dynamics simulations at a temperature where anharmonic effects are more important.

  2. The performance of the new approach is based on comparing its acceptance ratio to molecular dynamics simulations. How do actual computation times compare? How does the performance (computation time) scale with system size including the possibility to run calculation in parallel on many cores?

  3. Presentation. Title and abstract suggest application of the method to a wild variety of problems in solid state physics. However, such are mentioned only marginally in the conclusions while the main text fully focuses on the application to lattice dynamics. Experts in lattice dynamics may thus overlook this work and its relevance if not highlighted better. At several points the manuscript would profit from more quantitative statements.

For instance,

Lines 80-81: Limitations and applications should be discussed in more details.

Lines 102-103: Phase transitions are excluded by the "reasonable" class. This should me mentioned and the application of the approach to different kind of phase transitions could be addressed.

Lines 114-115: "too wild" and "very quick" should be quantified.

Lines 136- 137: "barely noticeable" and "hardly visible" obviously depend on how the data is plotted. Please quantify.

Figs. 1 and 2. correlations between $E_{k var}$ and $E_{p var}$ could be discussed.

Lines 186-189: Asymptotic production of target distribution for any non-vanishing prior distribution requires a citation.

Lines 213-214: Please explain why parameters are independent and their values not critical. Are there limitations?

Figures 5 and 6 should be discussed in more detail. Agreement and differences need to be pointed out. Fig. 5 is confusing. It's caption suggest that molecular dynamics was used to extract harmonic phonon frequencies, while the text states that higher order (anharmonic) force constants were extracted. It would be nice to compare harmonic phonon frequencies to both anharmonic phonon frequencies obtained from molecular dynamics and from the new method and discuss agreement and differences in detail for at least two different temperatures. Fig. 6 is lacking information on lifetimes of the acoustic branches with small momenta and small energies close to the $\Gamma$-point. It would be nice to discuss convergence and numerical limitations for these.

Both figures are very difficult to read because they are small and contain too much data. Splitting into sub-panels where the same number of samples are compared could help.

In summary, the presented approach is highly innovative and worth to be published but the presentation needs to be improved to make it convincing.

Requested changes

Please see report

  • validity: high
  • significance: high
  • originality: top
  • clarity: good
  • formatting: reasonable
  • grammar: reasonable

Author:  Paweł Jochym  on 2021-04-26  [id 1383]

(in reply to Report 2 by Bjorn Wehinger on 2021-03-10)
Category:
remark
answer to question

Reply to the report of Dr Wehinger

We would like to thank Dr Wehinger for careful reading of the manuscript and his positive opinion on our work.

The authors of the manuscript "High Efficiency Configuration Space Sampling – probing the distribution of available states" present a new method for studying lattice dynamics at finite temperatures. Their approach is based on configuration sampling the distribution of available states using the Metropolis-Hasting algorithm with a prior probability distribution derived from harmonic lattice dynamics. The authors compute anharmonic phonon dispersions and lifetimes for 3C-SiC to validate their approach and claim high computational performance due to a large observed acceptance ratio. The idea is highly original and I expect significant impact for the study of thermal properties at finite temperatures in large crystalline systems containing many atoms per unit cells. In order to fully convince the reader and justify publication in SciPost Physics, I recommend the authors to address and clarify the following:

We would like to point out that the presented approach does not depend on strict harmonicity of the system. The Eq. (4) and its description (l. 78-87) explicitly point to the impact of the anharmonicity on the formulas used in the proposed method. In particular, the normality of the distribution is not impacted - since it originates from the Central Limit Theorem (CLT, Eq. 5). What may be influenced is the value of the mean and the variance of the distribution - which will skew the temperature scale and possibly diminish the fidelity of our approximation of the thermal equilibrium state. Since both referees missed this point we have expanded our explanation of this issue to make it more clear to the reader.

Additionally, while we use lattice dynamics as an example in the text, the potential applicability of the proposed method is broader - it may be useful in other places where we need to reproduce the configuration of the system of atoms in thermal equilibrium in non-zero temperature. This fact is mentioned in the abstract but we will expand the conclusions by mentioning it there as well.

  1. The lattice dynamics of 3C-SiC at room temperature can be described fairly well by the harmonic approximation. It seems thus no big surprise that a prior probability distribution derived from harmonic lattice dynamics converges successfully and quickly. But how well does it work for a more anharmonic situation? Although the chosen potentials might not be accurate close to melting it would be a very nice illustration to compare the dispersions and lifetimes to molecular dynamics simulations at a temperature where anharmonic effects are more important.

Indeed, the chosen system is not strongly anharmonic at T=300K. But still there is enough anharmonicity in the model to produce 5ps phonon lifetimes plotted in Fig. 6. Also, the Tersoff potential selected for the study is not a simplistic, harmonic, two-body potential. It is a published, effective model of interactions in the Si-C compounds.

We would like to stress that the closeness of the prior distribution to the target (Figs 3 and 4) originates from the size of the system and careful selection of the prior generating algorithm (Eq. 6 and description in l. 172-183). As we noted in the reply to the first referee the extreme cases of anharmonicity dominating the right hand side of the Eq. 4 for all, or most coordinates may be beyond the direct applicability of the proposed method. To illustrate the point, we have added to Figures 3, 4, 5, and 6 the calculations performed for higher temperatures (up to T=2000K) closer to the melting point of 3C-SiC demonstrating effectiveness of the proposed approach even in high temperatures.

  1. The performance of the new approach is based on comparing its acceptance ratio to molecular dynamics simulations. How do actual computation times compare? How does the performance (computation time) scale with system size including the possibility to run calculation in parallel on many cores?

The computational cost is essentially proportional to the number of requested configurations plus necessary burn-in samples (1-10, can be limited to 1-2 with careful selection of initial displacement variation). Due to the details of the Metropolis-Hastings algorithm this cost is independent of the acceptance ratio. Low acceptance ratio leads to low quality of the generated distribution, not a direct increase in computational cost. This increase stems from the fact that with low acceptance more samples are required for the reasonable fidelity of the produced distribution. In comparison with MD calculations, each generated configuration is equivalent to one time step in trajectory. However, in case of the DFT-based calculations, the MD procedure can be optimized by starting each step from the charge density/wave functions converged in the previous step. Due to the fact that samples generated by HECSS are independent, this optimization is not easily available in DFT-based calculations. This amounts to approximately twice as many electronic SCF steps per evaluated configuration. Thus, n-configurations HECSS run is equivalent to approximately 2*(n+10) time steps of the MD calculation. In our experience this is not enough to provide even single, well-thermalized sample for n<500.

Regarding the parallel computation: In current implementation each configuration evaluation may be run on multiple cores but the sample generation is strictly serial. The near-independence of generated samples provides opportunity for future splitting of the computation to multiple processes. Naturally, each temperature scan may be run as a separate process with full linear scaling.

We will add analysis of the computational cost of the HECSS approach to the final paragraph of the text.

  1. Presentation. Title and abstract suggest application of the method to a wild variety of problems in solid state physics. However, such are mentioned only marginally in the conclusions while the main text fully focuses on the application to lattice dynamics. Experts in lattice dynamics may thus overlook this work and its relevance if not highlighted better. At several points the manuscript would profit from more quantitative statements.

We will expand the abstract to better reflect our focus - which is indeed, at this moment, on lattice dynamics applications. The other applications mentioned in the abstract are our suggestions of other fields where this type of procedure may be beneficial.

Lines 80-81: Limitations and applications should be discussed in more details. Lines 102-103: Phase transitions are excluded by the "reasonable" class. This should me mentioned and the application of the approach to different kind of phase transitions could be addressed.

Following the comment of the first referee we have expanded the description of probable limitations of the proposed method (phase transitions, highly anharmonic systems).

Lines 114-115: "too wild" and "very quick" should be quantified.

Lines 136- 137: "barely noticeable" and "hardly visible" obviously depend on how the data is plotted. Please quantify.

We have replaced these imprecise phrases with quantitative description showing the speed of convergence and cited the appropriate literature.

Figs. 1 and 2. correlations between E_kvar and E_pvar could be discussed.

The correlation between variances of the kinetic and potential distribution comes directly from energy conservation and statistical mechanics. Both energies are part of the Hamiltonian and sum up to the total energy. Thus, due to the energy conservation their variances should match. We have added the appropriate sentence to the discussion at the end of section 2.

Lines 186-189: Asymptotic production of target distribution for any non-vanishing prior distribution requires a citation.

Appropriate citation has been added to the list of references.

Lines 213-214: Please explain why parameters are independent and their values not critical. Are there limitations?

The independence from the system (supercell) size stems from the connection with the displacement distribution - it is our conclusion drawn from the experience gained during the development of the HECSS scheme. If the interactions are reproduced reasonably well in the small supercell (e.g. single crystallographic unit cell) the average size of thermal displacement is expected to be the same as in larger supercell due to the same energy per degree of freedom (i.e. temperature) and very similar shape of the potential. The independence and the practical ranges of the parameters cited in the text are derived from the multiple tests run during the development of the HECSS code. We have added a sentence explaining this property and rephrased the surrounding text to make this issue more clear to the reader.

Figures 5 and 6 should be discussed in more detail. Agreement and differences need to be pointed out. Fig. 5 is confusing. It's caption suggest that molecular dynamics was used to extract harmonic phonon frequencies, while the text states that higher order (anharmonic) force constants were extracted. It would be nice to compare harmonic phonon frequencies to both anharmonic phonon frequencies obtained from molecular dynamics and from the new method and discuss agreement and differences in detail for at least two different temperatures.

The phonon frequencies presented in Fig. 5 are derived by fitting of a third order anharmonic model to both datasets and the frequencies are derived from this model. The lifetimes from the Fig. 6 are obtained from the same model using ALAMODE to compute anharmonic self-energy and phonon lifetimes from the third order coefficients in the fit (using relaxation time approximation). We have corrected a misleading description of Figs 5 and 6 and expanded the description to make the point clear. We have also included the RMS differences between phonon frequencies derived by both methods.

Fig. 6 is lacking information on lifetimes of the acoustic branches with small momenta and small energies close to the Γ-point. It would be nice to discuss convergence and numerical limitations for these.

The access to the vicinity of the zone-center is limited by the supercell size used in the calculation. The closest point provided by the supercell used in the paper (5x5x5, 1000 atoms) and reciprocal space sampling grid (20x20x20) is located at 1/10 of the zone size from the center. All data between this point and the zone center are interpolated from the fitted force constant matrices in real space. We will add information about the reciprocal space sampling to the text. Additionally we have expanded the presented data to include more temperatures: 100K 600K and 2000K.

Both figures are very difficult to read because they are small and contain too much data. Splitting into sub-panels where the same number of samples are compared could help.

Figure 5 is intended to show small difference between frequencies computed from both data sets. We agree that the presentation in both Fig. 5 and 6 will benefit from such split and we have replaced both figures by separate panels containing data sets of the same size. We have also added additional temperatures - as mentioned above. The description has been modified appropriately.

In summary, the presented approach is highly innovative and worth to be published but the presentation needs to be improved to make it convincing.

We hope that the above explanations and corrections to the text make our paper convincing and clarify all the issues raised by Dr Wehinger.

Paweł T. Jochym, Jan Łażewski

---

## Round 2 · Referee Report · Bjorn Wehinger (Referee 2) · 2021-5-12

Report
Referee report for the revised manuscript entitled "High Efficiency Configuration Space Sampling – probing the distribution of available states".
The authors have carefully addressed all my comments in the revised version of the manuscript. The new version of the manuscript properly describes the new method and nicely illustrates its application and limitations to lattice dynamics calculations at finite temperatures. As I have mentioned in my first report, the idea is highly original and clearly opens a new pathway in the study of thermal properties at finite temperatures applicable to large systems that are difficult to address otherwise. The revised version of the manuscript fulfills all general acceptance criteria. I believe that it is of great interest to a broad audience and thus recommend publication in SciPost Physics.
I only have a few minor comments:
-
How does the applied temperature adjustment of 90K mentioned for the calculation of phonon life times at a temperature of 600K compare to an estimated correction obtained by "deriving the $C_n$ coefficients from the force constants matrices" as explained in Section 2?
-
Labels in Fig. 5 disagree with description in the figure caption.
We would like to thank Dr Wehinger for a quick response and his favourable outlook on our revised manuscript. We are glad that Dr Wehinger found our reply and corrections satisfactory and convincing and would like to immediately address his remaining comments:
1) How does the applied temperature adjustment of 90K mentioned for the calculation of phonon life times at a temperature of 600K compare to an estimated correction obtained by "deriving the $C_n$ coefficients from the force constants matrices" as explained in Section 2?
The temperature correction shown in Fig. 6 is intended only as an illustration that our hypothesis from section 2, that the leading correction to the proposed scheme is a temperature shift, is indeed plausible. The derivation of the correction term from the force-constant matrix requires developing of a self-consistent correction procedure based on the higher order (above quadratic) force constants matrices. We agree that this is a very interesting direction of the future research, but it is beyond the scope of the current paper which is intended to establish a baseline of the core HECSS method to be extended in the future.
2) Labels in Fig. 5 disagree with description in the figure caption.
This unfortunate mistake will be corrected on resubmission.
Paweł T. Jochym Jan Łażewski

Paweł Jochym on 2021-05-03 [id 1405]
The updated reply to the report of the first referee is included as a PDF attachment.
Attachment:
reply_1.pdf
Paweł Jochym on 2021-05-03 [id 1404]
The reply to the report of Dr Wehinger is included as a PDF attachment.
Attachment:
reply_2.pdf
Paweł Jochym on 2021-05-03 [id 1403]
Updated reply to the report of the First Referee
We thank the referee for reading our paper and spotting omissions and mistakes in our text. We hope that the following clarifications and corrections will enable full understanding of our approach. We address the comments paragraph by paragraph quoting the referee before our response.
The text was intended as a demonstration of the proposed method not as a reproduction or prediction of a particular experimental result. This may perhaps explain unfortunate omission of sampled temperature in the text. While the temperature can in fact be extracted from the average energy in Fig. 3 and 4, it should also be stated in the text explicitly. The sampling temperature for all data in first version of the text is 300 K, chosen as standard ambient temperature. In the resubmitted version of the text we have corrected this omission. Furthermore, following suggestions of both referees, we have extended the presented data to higher temperatures - up to 2000K. The temperatures are indicated in the text.
The shape of the energy distribution is not determined by harmonicity of the potentials but by the Central Limit Theorem and the size of the system. The difference between prior distribution (which comes from our approximation of the displacement distribution) and the target distribution does not stem from the anharmonicity of the potential but mainly from the fact that the displacements of the atoms in the crystal are not independent (as stated in our text). Please note that we have no direct access to the potential energy of the system - we can only specify the geometry and calculate the resulting energy instead of directly generating the target energy distribution from Fig. 3. The high acceptance ratio comes from our selection of the displacement distribution and tuning algorithm described in the paper. The Metropolis-Hastings (M-H) algorithm can generate a target distribution from any prior which is non-zero over the domain. However, the acceptance ratio may be very low if you fail to use the prior which is a good approximation of target distribution. It is a well-known fact in the numerical statistics community that the good selection of the prior distribution is the key to an effective use of probability distribution sampling algorithms. To further demonstrate the effectiveness of the algorithm for wide range of temperatures we have included results up to T=2000K. We have also extended explanation of the procedure used to generate Fig. 3 (which clearly has acceptance ratio below 80% claimed in the text for a typical values of delta). This figure was generated with artificially large value of delta (0.1 instead of 0.005-0.02) to make difference between prior and posterior distributions easily visible.
The issue of sample correlation would be indeed important if we used a 'random walk'-type algorithm for the prior generation (which is a popular variant of the M-H algorithm). Instead, we use independent samples and the only possible correlation between them arises from a very small change (less than 2%) in the variance of the position distribution. We discuss this issue in the paragraph starting in line 192 (212 in revised text). On the other hand, the MD derived data has obvious autocorrelations - note that we can derive phonon frequencies from the Fourier transform of the velocity autocorrelation function along the MD trajectory. Thus, it is necessary to separate sampling points on the trajectory by substantial intervals allowing for these correlations to die out. Nevertheless, all time steps between the sampling points still need to be calculated, which leads to large inefficiency of the MD as a configuration generator. To further clarify the issue we have expanded the explanation in the text in paragraph starting at line 212 (revised text).
Indeed, for the purely harmonic system the phonon frequency test is not useful since phonon frequencies are independent from the displacement size. However, if the system considered in the paper were close to harmonic, we would expect to obtain very long phonon lifetimes (since they are infinite in the harmonic system). The data in Fig. 6 demonstrates that many of the phonon modes exhibit lifetimes below 10ps - showing non-negligible anharmonicity in the model. This fact provides justification for the validity of the phonon frequency test. Furthermore, expanded temperature data of new Fig. 5 (up to 2000K), demonstrates that anharmonicity induced by high temperatures has some small influence on the convergence of phonon data but not on the converged results (lower row of Fig. 5) which shows good agreement between MD and HECSS data in full range of temperatures. Additionally, we have included in the text RMS errors for frequencies obtained with both methods. The phonon lifetimes are very sensitive to the accuracy of the model. This is especially true in case of large values which indicate small deviations from harmonicity and usually carry large error bars. Unfortunately, the large range (close to four orders of magnitude) of the values of lifetimes makes the simple RMS measure of differences very misleading - since the differences at high end of the range will dominate the sum. However, we agree that the previous Figure 6 was indeed not very clear. Thus, we have replaced it with the separate plot for three temperatures (T=100, 300, and 600K) splitting the small-sample data set to a separate row. We think that the new Fig.6 clearly demonstrates good agreement between data obtained with MD and HECSS procedure over 4 orders of magnitude in phonon lifetime.
The number of steps (30 000) used in the introduction was a typical relaxation time of a long-run MD suggested by the often used "rule of thumb" in MD calculations (50 times period of typical vibrations in the system). For 3C-SiC: $f\approx 10$THz = $10^{13}$Hz $ \Rightarrow t=10^{-13}$s = 100fs; 50 * 100fs = 5ps. With 1fs time step that equals 5000 steps minimum run where we can use at most half of it for actual data (you need to provide time to obtain thermal equilibrium). If we need approx. 30 data points (as required by anharmonic calculations, see Fig. 6) and they should be separated by at least 1ps interval (at least 10 typical vibrations) we get approximately 30 000 steps. The cited number itself has no 'magical' value and results from the setup of the calculations presented in the paper. To avoid impression that the number 30 000 has any special meaning, we have replaced the number by the phrase: "thousands of MD steps".
Eq. (1): we have added to the revised text a sentence introducing the missing symbols: $x_n$ - generalized coordinate or momentum, $H$ - Hamiltonian, $T$ - temperature, $k_B$ - Boltzmann constant, $\delta_{mn}$ - Kronecker delta.
Eq. (2): We make no two-body assumption, neither implied nor explicit. The formulation of equipartition theorem, Eq. (1), explicitly concerns single coordinates (the only non-zero term due to the Kronecker delta) and makes no assumption on the form of the Hamiltonian $H$. The Taylor expansion, Eq. (2), is not a complete expansion in all coordinates $q$ (note the scalar $q$ symbol). It is the Taylor expansion in a single coordinate with coefficients ($C_n$ - proportional to partial derivatives of energy with respect to this coordinate) which are functions of all the other coordinates in the system. What is more, the calculations presented in the paper use the mentioned Tersoff potential developed in refs 17, 18. We understand that due to the formulation of the surrounding text this may not be entirely clear and may confuse the reader. To avoid this, we have added a clarifying sentence below Eq. (2).
The unfortunate sentence 80-81 brings nothing of importance to the text. Thus we have removed it in the resubmitted version and reformulated the surrounding paragraph.
Variance of several quantities is indeed divergent in some phase transitions. In cases where the transition involves divergent heat capacity this includes energy variance. Thus, our phrase :"...all physically interesting cases..." was indeed wrong. The sentence has been corrected and we clearly state that in cases where energy variance diverges, the procedure cannot be used. We thank the referee for spotting this important fact.
The Eq. (5) was an attempt to formally write asymptotic relation of the Central Limit Theorem described in the paragraph 99-102. The CLT is indeed not a limit relation but asymptotic distribution convergence relation and the Eq. (5) should use appropriate notation for such relations as convergence in distribution:
"Single server" mentioned in line 125 was used as a rough indication of the computational effort involved in the described task. It is nothing out of ordinary: 2x4 cores CPU and 32GB RAM. This is actually a fairly under-powered and old machine, less powerful than some of newer generation laptops. The information has been added to the sentence in revised text.
In some cases the random number generation may be indeed fairly expensive but in the case of typical systems of tens of atoms, the energy and forces evaluation is much more time-consuming. For instance, the random number generator we have used (from SciPy.stats library) takes 180$\mu$s to generate 3000 random numbers required to create one sample for the 5x5x5 supercell of 3C-SiC. The single evaluation of energy for the same cell (1000 atoms) takes 4ms (20 times longer) using ASAP3 with OpenKIM model from our calculations. A more sophisticated interaction model is bound to be even more time-consuming. Furthermore, molecular dynamics requires calculating multiple time steps per every generated sample. Considering this facts, we maintain that the proposed HECSS approach offers substantial advantage over MD as a source of configuration data.
Paweł Jochym on 2021-04-30 [id 1400]
Reply to the report of Dr Wehinger
We would like to thank Dr Wehinger for careful reading of the manuscript and his positive opinion on our work.
We would like to point out that the presented approach does not depend on strict harmonicity of the system. The Eq. (4) and its description (l. 78-87) explicitly point to the impact of the anharmonicity on the formulas used in the proposed method. In particular, the normality of the distribution is not impacted - since it originates from the Central Limit Theorem (CLT, Eq. 5). What may be influenced is the value of the mean and the variance of the distribution - which will skew the temperature scale and possibly diminish the fidelity of our approximation of the thermal equilibrium state. Since both referees missed this point we have expanded our explanation of this issue to make it more clear to the reader.
Additionally, while we use lattice dynamics as an example in the text, the potential applicability of the proposed method is broader - it may be useful in other places where we need to reproduce the configuration of the system of atoms in thermal equilibrium in non-zero temperature. This fact is mentioned in the abstract but we will expand the conclusions by mentioning it there as well.
Indeed, the chosen system is not strongly anharmonic at T=300K. But still there is enough anharmonicity in the model to produce 5ps phonon lifetimes plotted in Fig. 6. Also, the Tersoff potential selected for the study is not a simplistic, harmonic, two-body potential. It is a published, effective model of interactions in the Si-C compounds.
We would like to stress that the closeness of the prior distribution to the target (Figs 3 and 4) originates from the size of the system and careful selection of the prior generating algorithm (Eq. 6 and description in l. 172-183). As we noted in the reply to the first referee the extreme cases of anharmonicity dominating the right hand side of the Eq. 4 for all, or most coordinates may be beyond the direct applicability of the proposed method. To illustrate the point, we have added to Figures 3, 4, 5, and 6 the calculations performed for higher temperatures (up to T=2000K) closer to the melting point of 3C-SiC demonstrating effectiveness of the proposed approach even in high temperatures.
The computational cost is essentially proportional to the number of requested configurations plus necessary burn-in samples (1-10, can be limited to 1-2 with careful selection of initial displacement variation). Due to the details of the Metropolis-Hastings algorithm this cost is independent of the acceptance ratio. Low acceptance ratio leads to low quality of the generated distribution, not a direct increase in computational cost. This increase stems from the fact that with low acceptance more samples are required for the reasonable fidelity of the produced distribution. In comparison with MD calculations, each generated configuration is equivalent to one time step in trajectory. However, in case of the DFT-based calculations, the MD procedure can be optimized by starting each step from the charge density/wave functions converged in the previous step. Due to the fact that samples generated by HECSS are independent, this optimization is not easily available in DFT-based calculations. This amounts to approximately twice as many electronic SCF steps per evaluated configuration. Thus, n-configurations HECSS run is equivalent to approximately 2*(n+10) time steps of the MD calculation. In our experience this is not enough to provide even single, well-thermalized sample for n<500.
Regarding the parallel computation: In current implementation each configuration evaluation may be run on multiple cores but the sample generation is strictly serial. The near-independence of generated samples provides opportunity for future splitting of the computation to multiple processes. Naturally, each temperature scan may be run as a separate process with full linear scaling.
We will add analysis of the computational cost of the HECSS approach to the final paragraph of the text.
We will expand the abstract to better reflect our focus - which is indeed, at this moment, on lattice dynamics applications. The other applications mentioned in the abstract are our suggestions of other fields where this type of procedure may be beneficial.
Following the comment of the first referee we have expanded the description of probable limitations of the proposed method (phase transitions, highly anharmonic systems).
We have replaced these imprecise phrases with quantitative description showing the speed of convergence and cited the appropriate literature.
The correlation between variances of the kinetic and potential distribution comes directly from energy conservation and statistical mechanics. Both energies are part of the Hamiltonian and sum up to the total energy. Thus, due to the energy conservation their variances should match. We have added the appropriate sentence to the discussion at the end of section 2.
Appropriate citation has been added to the list of references.
The independence from the system (supercell) size stems from the connection with the displacement distribution - it is our conclusion drawn from the experience gained during the development of the HECSS scheme. If the interactions are reproduced reasonably well in the small supercell (e.g. single crystallographic unit cell) the average size of thermal displacement is expected to be the same as in larger supercell due to the same energy per degree of freedom (i.e. temperature) and very similar shape of the potential. The independence and the practical ranges of the parameters cited in the text are derived from the multiple tests run during the development of the HECSS code. We have added a sentence explaining this property and rephrased the surrounding text to make this issue more clear to the reader.
The phonon frequencies presented in Fig. 5 are derived by fitting of a third order anharmonic model to both datasets and the frequencies are derived from this model. The lifetimes from the Fig. 6 are obtained from the same model using ALAMODE to compute anharmonic self-energy and phonon lifetimes from the third order coefficients in the fit (using relaxation time approximation). We have corrected a misleading description of Figs 5 and 6 and expanded the description to make the point clear. We have also included the RMS differences between phonon frequencies derived by both methods.
The access to the vicinity of the zone-center is limited by the supercell size used in the calculation. The closest point provided by the supercell used in the paper (5x5x5, 1000 atoms) and reciprocal space sampling grid (20x20x20) is located at 1/10 of the zone size from the center. All data between this point and the zone center are interpolated from the fitted force constant matrices in real space. We will add information about the reciprocal space sampling to the text. Additionally we have expanded the presented data to include more temperatures: 100K 600K and 2000K.
Figure 5 is intended to show small difference between frequencies computed from both data sets. We agree that the presentation in both Fig. 5 and 6 will benefit from such split and we have replaced both figures by separate panels containing data sets of the same size. We have also added additional temperatures - as mentioned above. The description has been modified appropriately.
We hope that the above explanations and corrections to the text make our paper convincing and clarify all the issues raised by Dr Wehinger.
Paweł T. Jochym, Jan Łażewski

---

## Round 2 · Referee Report · Anonymous (Referee 1) · 2021-5-19

Report
The scope and current limitations of the HECSS scheme are now clearly presented. All the previously missing details required for reproducibility are now reported in the manuscript.
I appreciate that Fig. 5 and 6 show the increasing loss of accuracy of the current version of the HECSS as the temperature is increased.
I agree with the authors that high-temperature properties are challenging also using existing MD schemes.
Even if the CLT holds, i.e., even if the variance is finite, the convergence to the expected normal distribution can get rather slow.
Perhaps this can be better understood in terms of the tails of the potential energy distribution --- not clearly visible in Fig. 2 --- which get "heavier" at high-temperature because of the anharmonicity.
Investigating the role of these tails, e.g., perturbatively, may reveal a strategy to improve the agreement between the HECSS and MD without resorting to a somehow arbitrary "temperature calibration".

---

## Round 2 · Author Response

Your reference: scipost_202101_00011v1
Corresponding author: Paweł T. Jochym Address: Institute of Nuclear Physics, Radzikowskiego 152, 31-342 Cracow, Poland email: pawel.jochym@ifj.edu.pl
Title: High Efficiency Configuration Space Sampling - probing the distribution of available states
Authors: Paweł T. Jochym and Jan Łażewski Type: regular article
Dear Dr Attaccalite,
Thank you for arranging the review of our paper. We are glad that criticism of both referees contributed to the improvement of quality of our work. In fact, both referees noticed strong aspects of our idea and highlighted weak points of the presentation causing possible confusion of the reader. None of the key assumptions of the method have been questioned. The second referee went even further and in his summary rating recognized our approach as of the "high validity and significance" and "top originality".
We thank both referees for careful reading of our text and their valuable remarks. We regret a few mistakes and some deficiencies in presentation pointed by the referees. Following their advice we have made substantial revision of the text and figures correcting all mistakes and omissions as well as extending the explanations to make the presentation clearer.
Please find included a detailed response to both referees. We have already submitted an early response to the first review. This initial response is still valid, but we have reformulated it to closely reflect changes we have made in the text.
We have addressed all points raised by the referees, applied all their suggestions and answered all questions. We believe that after these corrections the manuscript is ready for publication in SciPost Physics without delay.
This resubmission includes: - the revised text, - the detailed reply to both referees (submitted as reply on the submission page), - list of changes
Sincerely Yours, Paweł T. Jochym Jan Łażewski

---

## Round 2 · List of Changes

Summary of changes in scipost_202101_00011v1/Jochym&Lazewski:
In response to the referee reports we made the following changes in the manuscript:
The Eq. 5 has been corrected.
The definition of the method was supplemented and the scope of its applicability was more precisely specified.
Calculations are extended up to 2000 K.
Figures 3, 4, 5 and 6 were split to several panels to separate data for different temperatures and increase readability of the contents.
Imprecise statements in our presentation (like "too wild", "very quick", "hardly visible" etc.) were quantified.
We have modified the text as recommended by the referees - these changes are described in the replies.
Minor modifications of the text were made in a few other places in order to improve the reading and to remove some typographical errors.

---

## Round 3 · Author Response

Following the comments of the referees and editor request we have corrected the mistake in the caption of the Fig. 5 and added a sentence indicating promising future direction of development for the proposed method.

---

## Round 3 · List of Changes

• Caption of the Fig. 5 corrected
  • Sentence on future research direction added

---

## Editorial Decision

published